# Ex Vivo Drug Sensitivity Correlates with Clinical Response and Supports Personalized Therapy in Pediatric AML

**DOI:** 10.3390/cancers14246240

**Published:** 2022-12-18

**Authors:** Debbie C. Strachan, Christine J. Gu, Ryosuke Kita, Erica K. Anderson, Michelle A. Richardson, George Yam, Graham Pimm, Jordan Roselli, Alyssa Schweickert, Maci Terrell, Raushan Rashid, Alan K. Gonzalez, Hailey H. Oviedo, Michelle C. Alozie, Tamilini Ilangovan, Andrea N. Marcogliese, Hiroomi Tada, Marianne T. Santaguida, Alexandra M. Stevens

**Affiliations:** 1Notable Labs, Foster City, CA 94404, USA; 2Division of Pediatric Hematology/Oncology, Department of Pediatrics, Baylor College of Medicine, Houston, TX 77030, USA; 3Department of Pathology, Baylor College of Medicine, Houston, TX 77030, USA

**Keywords:** pediatric acute myeloid leukemia, precision medicine, flow cytometry, ex vivo drug sensitivity, combination therapy, personalized medicine, ADE, bortezomib, panobinostat

## Abstract

**Simple Summary:**

Children with acute myeloid leukemia (AML) experience unacceptably poor survival outcomes and are at high risk of relapse. Current treatment options are limited, and standard strategies rely on intensive chemotherapy to achieve remission, frequently resulting in treatment-related morbidities and significant late adverse effects. The use of an ex vivo drug sensitivity platform has potential clinical utility to aid individualized patient risk assignment, and it could allow for personalized treatment regimens, including identifying novel therapies for patients who are identified to be at a very high risk of treatment failure. In this study, we show that ex vivo drug sensitivity correlates with clinical response measures in a cohort of children with AML who received conventional chemotherapy. We also demonstrate preferential sensitivity ex vivo between conventional chemotherapy and the combination of bortezomib and panobinostat in a subset of patient samples. Our results support the value of an ex vivo drug sensitivity platform to identify individualized precision therapy for children with AML.

**Abstract:**

Acute myeloid leukemia (AML) is a heterogeneous disease that accounts for ~20% of all childhood leukemias, and more than 40% of children with AML relapse within three years of diagnosis. Although recent efforts have focused on developing a precise medicine-based approach towards treating AML in adults, there remains a critical gap in therapies designed specifically for children. Here, we present ex vivo drug sensitivity profiles for children with de novo AML using an automated flow cytometry platform. Fresh diagnostic blood or bone marrow aspirate samples were screened for sensitivity in response to 78 dose conditions by measuring the reduction in leukemic blasts relative to the control. In pediatric patients treated with conventional chemotherapy, comprising cytarabine, daunorubicin and etoposide (ADE), ex vivo drug sensitivity results correlated with minimal residual disease (r = 0.63) and one year relapse-free survival (r = 0.70; AUROC = 0.94). In the de novo ADE analysis cohort of 13 patients, AML cells showed greater sensitivity to bortezomib/panobinostat compared with ADE, and comparable sensitivity between venetoclax/azacitidine and ADE ex vivo. Two patients showed a differential response between ADE and bortezomib/panobinostat, thus supporting the incorporation of ex vivo drug sensitivity testing in clinical trials to further evaluate the predictive utility of this platform in children with AML.

## 1. Introduction

Pediatric acute myeloid leukemia (AML) is a rare and heterogeneous disease, with roughly 600 cases diagnosed in the United States each year [1,2,3]. Although complete remission (CR) rates are ~90%, there has been minimal improvement in clinical outcomes for children with AML over the past few decades [4,5,6]. Event-free survival (EFS) in pediatric AML is still at ~45%, and overall survival (OS) is at ~65% after three years, with relapse occurring in almost half of children diagnosed with AML [4,5,7]. Children at the highest risk of relapse are those with poor genetic features and those who require bone marrow transplants to have the best chance of being cured. Even with this intensification of therapy, only one in three children who proceed to have a transplant survive after three years [4].

Standard therapy for children with AML continues to rely on high-dose chemotherapy regimens developed in the 1970s; these include cytarabine and anthracyclines to induce remissions, with few options for relapsed or refractory periods [8]. Past efforts to improve survival in pediatric AML have focused on increasing the dose intensity of chemotherapy. Although this approach has led to moderate improvement in survival, relapse remains a frequent problem, and intensive chemotherapy carries a serious risk of toxicity, including infection, with ~20% of childhood survivors of pediatric AML having significant cardiac dysfunction later in life [9].

Since 2017, targeted therapies, including venetoclax, FLT3 inhibitors, and IDH inhibitors, have been approved for older adults with AML (age 60 years and older) [10], and there is the assumption that these treatments can also be applied to children with AML. A comparison of findings from the Therapeutically Applicable Research to Generate Effective Treatments (TARGET) AML initiative, a collaborative Children’s Oncology Group (COG)—National Cancer Institute (NCI) project, underscore both pediatric and adult AML as two separate diseases with distinct genomic features and pathophysiologies [11,12]. Given the poor outcomes in pediatric AML, there is an urgent need for more effective and less toxic therapies that are designed specifically for children with AML.

To this end, precision medicine holds promise in terms of matching patients to the right therapies through cytogenetics and sequencing. In the NCI-COG Pediatric MATCH (Molecular Analysis for Therapy Choice) study, actionable mutations were identified in 31% of tumors sequenced with treatment arm assignment and enrollment, in 28% and 13% of patients screened, respectively [13]. In the first phase 2 treatment trial to be completed as part of the MATCH study, selumetinib, a mitogen-activated protein kinase (MAPK) kinase inhibitor, showed limited efficacy in 20 patients harboring active mutations in the MAPK pathway [14]. This finding indicates that pathway mutation status and genetic approaches alone may be insufficient for predicting treatment responses. The ongoing Intergroup LEAP (Less Intense AML Therapy Platform) trial for leukemia is testing a rolling arm “cassette” design where patients are stratified by FLT3 mutation status, in addition to age and performance status, and randomized into treatment arms [15,16].

We, along with others, have shown the feasibility and clinical utility of functional precision medicine assays in hematologic malignancies [16,17,18,19,20,21,22,23,24,25,26]. Historically, ex vivo predictive analytics involve clinical samples and a single assay, often a bulk viability assay, that is not specific to tumor cells or to a relevant biologic response. By contrast, the flow cytometry-based precision medicine platform we developed is unique in terms of its ability to measure both cell phenotype and function on a particular scale. We previously demonstrated that ex vivo drug sensitivity can predict clinical responses in myelodysplastic syndrome [19]. In a cohort of 21 patients, results from our ex vivo drug sensitivity screening had a positive predictive value of 0.92, and a negative predictive value of 0.82, with an overall accuracy of 0.85. This functional precision medicine platform has the potential to tailor the treatment plans for each patient, both at disease onset and relapse.

In this study, we evaluated the use of an ex vivo drug sensitivity platform (DSP) to predict clinical response, identify potential novel drug combinations, and inform personalized therapies for children with AML. Using a custom robotic flow cytometry-based platform, diagnostic patient samples were assayed for sensitivity in response to a maximum of 78 dose conditions. In a cohort of 13 de novo pediatric AML patients that received conventional chemotherapy comprising of cytarabine, daunorubicin, and etoposide (ADE), DSP results for ADE were compared with minimal residual disease status at the end of induction and after one year of relapse-free survival. Within the 78 dose conditions tested in the DSP, 31 unique drugs were included in single agent and combination settings to identify potential drug combinations that show comparable or greater sensitivity to ADE ex vivo.

## 2. Materials and Methods

### 2.1. Patients

All patients were diagnosed at Texas Children’s Hospital (Houston, TX, USA) between May 2015 and October 2020, and they received treatment in accordance with COG protocol AAML0531 or AAML1031. Cytogenetics and next-generation sequencing were conducted as part of their clinical care, as previously described [4,27,28]. Risk stratification was conducted as per protocol AAML1831 guidelines. All patients or guardians provided informed consent in accordance with the Declaration of Helsinki, and subjects were enrolled on a protocol approved by the Institutional Review Board of Baylor College of Medicine.

### 2.2. Sample Collection

Peripheral blood or bone marrow aspirate samples were collected at Texas Children’s Hospital at various clinical time points and shipped overnight to Notable Labs (Foster City, CA, USA). Diagnostic samples were collected prior to the initiation of systemic treatment.

### 2.3. Ex Vivo Drug Sensitivity Platform

Ex vivo drug sensitivity testing was performed at Notable Labs (Foster City, CA, USA), using a custom robotic flow cytometry platform. Within 72 h of collection, fresh blood or bone marrow aspirate samples were incubated with a 1X RBC lysis buffer (eBioscience/Applied Biosystems, Santa Clara, CA, USA) to remove red blood cells. Cell pellets were resuspended in a StemSpan Serum-Free Expansion Medium II (SFEMII; STEMCELL Technologies, Vancouver, Canada) containing an HS-5 conditioned medium and Penicillin/Streptomycin (Corning, Corning, NY, USA), along with the following cytokines: rhIL-7, rhG-CSF, rhFLT3L, rhSCF, rhTPO (Miltenyi Biotec, Bergisch Gladbach, North Rhine-Westphalia, Germany); rhGM-CSF, rhIL-3, rhIL-6 (R&D Systems/Bio-Techne, Minneapolis, MN, USA). Resuspended cells were transferred to 384-well plates, with 15,000 cells per well, and they were incubated with the drug in triplicate, or with a DMSO control in sextuplicate (day 0). After 3 or 7 days following the addition of the drug, changes in tumor blast populations were assayed using flow cytometry. Patient samples with low total counts were assayed at a single time point on day 3 or day 7. Ex vivo drug sensitivity was calculated based on the number of blasts remaining for each condition compared with the DMSO control. For day 7 readouts, cytokines were replenished on day 3. For the HS-5 conditioned medium, HS-5 cells were seeded in a DMEM high glucose medium (ATCC) containing 10% fetal bovine serum (FBS; Gibco/Thermo Fisher Scientific, Waltham, MA, USA). After 48 h, or when cells reached 50–60% confluency, the medium was replaced with a SFEMII medium. After 72 h, the HS-5 conditioned medium was harvested.

### 2.4. Flow Cytometry and Blast Gating

Cells were incubated with antibodies for 20 min at 4 °C and analyzed on an iQue Plus flow cytometer. Several panels of antibodies were used, including: CD19 PACBLUE SJ25C1, CD3 PACBLUE HIT3A, CD38 BV785 HIT2, Annexin V PE (BioLegend, San Diego, CA, USA); CD33 BV510 WM53, CD34 BV605 563, CD11B APC-CY7 ICRF44, CD56 BV605 NCAM16.2 (BD Biosciences, Franklin Lakes, NJ, USA); CD45 FITC 2D1, CD14 PE 61D3, CD66B PE-CY7 G10F5, HLA-DR APC LN3, CD117 APC 104D2 (eBioscience); and DAPI (Sigma/Merck Group, St Louis, MO, USA). Live cells were defined as negative for DAPI or negative for both DAPI and Annexin V. The total blast population was defined as being CD45+ LIN- (CD3-/CD19-, CD66b-), and it was positive for at least one of the following blast markers: CD34, CD33, HLA-DR, CD117 or CD56 (Appendix A).

## 3. Data Analysis

Criteria for the exclusion of patient screenings from the DSP include technical errors relating to equipment malfunction and/or insufficient blast counts (<150 in DMSO control conditions). Individual wells with a population count greater than a threshold percentage that was 20–50% away from the median replicate value were excluded from analysis; thus, there was a higher tolerance for smaller population median values.

Computational analyses were performed in Python v.3.7.3, unless otherwise noted. Survival duration and the median length of follow-up were calculated on 22 February 2022. Normalized blast counts were calculated for each patient and DSP condition by dividing the mean blast count in the treated condition by the mean blast count for the DMSO control. Log odds ratios were calculated using the Haldane–Anscombe correction in R. GraphPad Prism (version 9; GraphPad Software, La Jolla, CA, USA), which was used to generate Kaplan–Meier survival curves. Principal component analysis was performed using the scikit-learn library for dose conditions across the cohort. Conditions with missing values were excluded from analysis.

## 4. Results

### 4.1. Thirty-One Pediatric AML Patients Were Profiled Using an Ex Vivo Drug Sensitivity Platform

Fresh blood or bone marrow aspirate samples were collected for this study from 31 pediatric AML patients (Figure 1A). For their first induction regimen, twenty-seven patients received ADE, three patients received liposomal daunorubicin and cytarabine (CPX-351), or a combination of daunorubicin, cytarabine, gemtuzumab, and atovaquone; the treatment for one patient was unknown (Figure 1B). Twenty-one out of the twenty-seven patients that received ADE were enrolled in NCT03568994, A Trial of Atovaquone combined with Conventional Chemotherapy (ATACC) for de novo AML in children, adolescents, and young adults. Atovaquone was given to twenty out of twenty-one ATACC patients, and for the purpose of this study, these patients are considered part of the ADE cohort.

The baseline patient characteristics and treatments are summarized in Table 1. Of the 27 patients that received ADE and from whom samples were collected, 24 had de novo disease (Figure 1B), and within this subset, 12 out of 24 (50%) patients were considered high risk in accordance with AAML1831 criteria. Of the twenty-four patients, five patients had FLT3-ITD mutations, and five non-overlapping patients had KMT2A rearrangements. Eleven out of twenty-four (46%) patients had M1/M2 histology, ten out of twenty-four (42%) were M4/M5, three out of twenty-four (13%) were M7 with a median age of 11 years (range, 0.6–19 years), and twelve out of twenty-four (50%) patients were female.

Diagnostic samples were collected prior to treatment from 22 out of 24 de novo AML patients that received ADE. Bone marrow aspirates comprised 10 out of 22 (45%) of diagnostic samples from de novo AML patients with the remaining 12 samples being peripheral blood specimens, with a high concordance in ex vivo drug sensitivity values between these two specimen types, as previously shown [19].

A minimum of 2 mL of blood or bone marrow aspirate was collected from each patient for the DSP. They were screened against 78 dose conditions, including 31 drugs and 47 drug combinations, using an automated flow cytometry-based platform (Figure 1C, Appendix A). Out of 31 patients, samples from at least one timepoint for 27 patients were successfully assayed with samples from the remaining four patients, and failures only occurred due to insufficient cell counts and/or instrumentation errors. Altogether, 48 samples from 27 patients were profiled in the DSP and compiled with patient metadata (Figure 1D). Two of these samples were from a patient-derived mouse xenograft (PDX) model for patient pAML17; however, no diagnostic specimen was collected for this patient, and we could not compare DSP results from fresh patient samples with PDX-derived samples.

### 4.2. Ex Vivo Drug Sensitivity in Response to ADE Correlates with Clinical Response

Ex vivo drug sensitivity to ADE was profiled for 13 out of 22 patients that had de novo disease and received ADE (Figure 1B). Day 3 DSP results for ADE were used for this study as this readout day included more patient samples than day 7. Within this de novo ADE analysis cohort (*n* = 13), we observed a correlation between ex vivo drug sensitivity in response to ADE and minimal residual disease (MRD) at the end of induction (EOI1), with an r = 0.63 (Figure 2A). Ten out of thirteen patients with MRD ≤ 1% at EOI1 showed a reduction in blasts in the DSP with ADE compared with the control. The remaining three patients with MRD > 1% showed less sensitivity to ADE ex vivo and had higher normalized blast counts compared with the MRD ≤ 1% group. The correlation with MRD was highest when the ADE combination condition was used, as compared with the use of single agents in the DSP (Appendix A). A correlation between ex vivo drug sensitivity to ADE and one year relapse-free survival (RFS) was also observed with r = 0.70 and an AUROC = 0.94 (Figure 2B).

Log odds ratios were used to measure associations between MRD and ex vivo DSP results and clinical attributes. A normalized blast score of 0.7 was applied to distinguish between high (DSP ≤ 0.7) and low sensitivity (DSP > 0.7) to ADE ex vivo. A normalized blast count in the DSP > 0.7 (indicating low sensitivity to ADE ex vivo) demonstrated increased odds of the patient having an MRD > 1% compared with any single genetic feature (FLT3-ITD, NPM1 mutation, KMT2A rearrangement, or IDH1 mutation) or clinical attribute (female, older than 13 years old, high risk) queried (Figure 2C).

Of the 13 patients in this cohort, patient pAML7 died of treatment-related mortality and was censored from relapse-free survival analysis. Moreover, one patient (pAML3) died of disease 17 months following diagnosis (Figure 2D). Patients were divided into two groups based on DSP with eight out of twelve (67%) patients having normalized blast counts ≤ 0.7 (high sensitivity) compared with four out of twelve (33%) patients with normalized blast counts > 0.7 (low sensitivity); patient pAML3 belonged to the low sensitivity group. Two out of twelve patients relapsed with a median follow-up length of 21 months (range, 17–31 months), and both patients showed a low sensitivity to ADE ex vivo with a DSP > 0.7 (Figure 2E).

### 4.3. Patient pAML3 Non-Responder and Patient pAML8 Responder Captures the Range of High and Low Drug Sensitivity across Multiple Conditions Tested Ex Vivo

Patients pAML3 and pAML8 both received ADE plus atovaquone for induction chemotherapy. Patient pAML3 failed induction chemotherapy whereas patient pAML8 had no measurable residual disease at the end of induction bone marrow evaluation (Figure 3A). Patient pAML3 was a 16 year-old male diagnosed with the high-risk AML M5a subtype, with cytogenetics significant for MLL-MLLT4 (KMT2A-AFDN) fusion. This patient relapsed after a bone marrow transplant following the successful reinduction of chemotherapy and ultimately died of disease 17 months following diagnosis. Patient pAML8 was a two year-old male diagnosed with the AML M7 subtype, and cytogenetics and molecular testing were significant for trisomy 10 and a WT1 mutation. As this patient was MRD negative at the end of induction, he was classified as being at a low risk of relapse per AAML1831 criteria, and the patient proceeded with chemotherapy alone. Patient pAML8 remains disease-free 22 months following induction chemotherapy.

Ex vivo drug sensitivity profiles for patients pAML3 and pAML8 were compared across 29 matched conditions (Figure 3B). Applying DSP > 0.7 as a reference for low or no sensitivity, neither patient’s leukemia cells were sensitive to 15 out of 29 (52%) conditions, but both were sensitive to four conditions that included cytarabine or etoposide. As indicated via the shading in the lower right quadrant, the pAML8 sample was sensitive to 10 conditions, including ADE, whereas the pAML3 sample was not sensitive.

We noted that the pAML8 sample was most sensitive to bortezomib in combination with panobinostat, and we evaluated other patient samples’ sensitivity to this combination compared with their sensitivity to ADE (Figure 3C). The combination of bortezomib/panobinostat showed greater efficacy ex vivo compared with single agents (Appendix A), and it had a greater additive effect in 3 AML cell lines (Appendix A).

A separation of patient samples into two distinct high and low sensitivity groups was most pronounced with bortezomib/panobinostat, whereas ADE, along with matched single agents, showed a more graded distribution across the 13 patients tested. For most of the distributions profiled, the range of ex vivo drug sensitivity was anchored by patient pAML3 (non-responder) on the low sensitivity end and patient pAML8 (responder) on the high sensitivity end.

### 4.4. Bortezomib in Combination with Panobinostat Shows the Highest Median Sensitivity out of the 37 Conditions Tested in the DSP

Up to 37 conditions were tested in the DSP for this cohort, and among these, bortezomib/panobinostat had the lowest median normalized blast count (DSP < 0.1) and showed the highest sensitivity (Figure 4A). Cytarabine, etoposide, and/or daunorubicin accounted for eight out of eleven of the most sensitive drug conditions in the DSP, and venetoclax, in combination with azacitidine, a hypomethylating agent (HMA), showed a comparable sensitivity to ADE ex vivo.

The range and distribution of normalized blast counts varied across conditions, and bortezomib/panobinostat showed the widest range and greatest separation between high and low sensitivity in the DSP (Figure 4B). Venetoclax, in combination with azacitidine, showed a narrower range for normalized blast counts (median 0.25, range 0.1–0.6), and it was the only condition for which all patients in the de novo ADE cohort showed a reduction in leukemic blasts ex vivo.

Principal component analysis (PCA) was performed to identify the clustering of conditions that could account for the differences observed in ex vivo drug sensitivity. Sixty percent of the variations in the DSP could be explained by PC1 (principal component 1) and PC2 (principal component 2); these components accounted for <15% of the observed variations (Figure 4C). Using a PC1 vs. PC2 PCA plot, drug combinations with cytarabine (including ADE single agents and combination, DA, cytarabine/etoposide) were found to cluster together and away from other conditions in the DSP, thus indicating that similar drug sensitivity patterns are shared in this cytarabine combination group. Bortezomib, in combination with panobinostat, was found to be the most different from all other conditions tested in the DSP, thus indicating the singularity of this combination when accounting for the differences in ex vivo drug sensitivity patterns.

### 4.5. Preferential Sensitivity between ADE and Bortezomib/Panobinostat Is Observed in a Subset of Pediatric AML Patients in the DSP

In comparison to patients pAML3 and pAML8, who showed similar responses to ADE and bortezomib/panobinostat in the DSP, patients pAML4 and pAML6 showed a different response in the DSP between these two conditions (Figure 5A). Patient pAML4 showed sensitivity to ADE, but not to bortezomib/panobinostat ex vivo. Conversely, patient pAML6 showed sensitivity to bortezomib/panobinostat, but not to ADE in the DSP; this patient also failed induction and relapsed (Table 1). Similar to patient pAML3, patient pAML1 also showed reduced sensitivity to both bortezomib/panobinostat and ADE, and this patient also failed induction.

Expanding this comparison to include venetoclax in combination with the HMAs (azacitidine and decitabine (Figure 5A,B)), color-mapping was applied to compare high and low ex vivo sensitivities across patients for each of the four conditions (Figure 5B). Eight patients had ex vivo drug sensitivity results for all four conditions. Within this subset, patient pAML7 showed a greater sensitivity to bortezomib/panobinostat and relatively lower sensitivities to ADE and venetoclax/HMA in the DSP. Patient pAML6, who showed a preferential sensitivity to bortezomib/panobinostat and not ADE, also showed greater sensitivity to venetoclax/HMAs compared with other patients in this cohort.

## 5. Discussion

These results support the idea that a clinical response in pediatric AML patients may be predicted using an ex vivo drug sensitivity platform. In contrast to bulk cytotoxicity assays, the DSP utilizes flow cytometry to directly measure changes in blast populations. We show that in pediatric AML patients treated with ADE, DSP results correlate with both MRD and one year RFS. Additionally, DSP results show a greater likelihood of predicting induction failure compared with any of the four single mutations or three clinical attributes queried. Patients pAML1, pAML3, and pAML6 showed low sensitivity for ADE ex vivo and all three patients failed induction, with relapse occurring in patients pAML3 and pAML6.

In addition to offering predictive value, the DSP also has the potential to provide suggestions on personalized therapy; this is because preferential sensitivity between ADE and bortezomib/panobinostat was observed in a subset of patients. This supports the use of an ex vivo drug sensitivity platform to inform and potentially identify personalized treatment options for pediatric AML patients. Of particular importance is the screening capability of the DSP to not only test drug sensitivity to conventional chemotherapy but also to compare sensitivity with other potential therapies for each patient. Using a custom robotics platform, up to 78 drug conditions can be tested from 2 mL of peripheral blood or bone marrow aspirate. This includes drug combination testing and analysis of combination activity to potentially identify more efficacious therapies and evaluate novel therapeutic strategies for children with AML. An intriguing option for this approach could be in selecting therapies for children with relapsed disease, or for those who have poor predicted outcomes with conventional therapy.

In the majority of patient samples screened, bortezomib, in combination with panobinostat, was found to have greater efficacy ex vivo compared with ADE. Bortezomib, a proteasome inhibitor, and panobinostat, a histone deacetylase inhibitor, have been shown to be safely administered with chemotherapy in children with AML [4,29]. Although the addition of bortezomib to conventional chemotherapy did not improve survival in pediatric AML patients [4], the combination of bortezomib and panobinostat has not been evaluated in children with AML.

Venetoclax, in combination with HMAs, was also observed to have comparable efficacy to ADE ex vivo. All eight patient samples in the ADE analysis cohort that were screened for sensitivity to venetoclax/azacitidine ex vivo showed a reduction in blasts compared with the control. In a single-center study, the combination of venetoclax and azacitidine was found to be well-tolerated in children with relapsed refractory AML [30], and larger studies are underway to better quantify the efficacy of this combination in pediatric AML. Beyond the use of the combination for relapsed and refractory disease, or as a bridge to transplant, venetoclax has been evaluated as a part of an upfront therapy treatment in pediatrics [31], and in adults, it has been used as a potential maintenance regimen after a hematopoietic stem cell transplant (“VIALE-M”, NCT04161885) [32]. Predicting responses for children for whom azacitidine and venetoclax are considered viable treatments may be of clinical utility.

An ex vivo drug sensitivity platform also has the potential to support individualized patient risk and to provide better treatment arm assignment in children with AML. The platform may allow for the modification of treatment and the identification of targeted therapies for very high-risk patients. Furthermore, a validated DSP may allow for the better stratification of high- versus low-risk patients to minimize the use of bone marrow transplantation and unnecessarily toxic therapies for lower risk patient subsets who are likely to have good outcomes.

Turnaround time from patient sample collection to DSP report generation is within a potentially clinically actionable timeframe, with a median of 15 days (range, 13–24 days) [19], and it is comparable to the 1–2 week turnaround time for genomic profiling. Faster turnaround times for the DSP that are closer to 1 week are currently being developed using custom software and robotics. This would allow for the addition of a targeted agent on day 11 following the completion of conventional chemotherapy; this is similar to what currently takes place as part of standard treatment for children with FLT3-ITD mutations.

Following the completion of this study, we have since expanded the platform to measure changes in other cell types to enable more complex analyses for profiling drug-specific changes in the patient’s tumor microenvironment. These efforts are intended to help advance our understanding of AML biology in children, as well as potentially enhance the predictive value and types of personalized therapies that can be evaluated using this platform.

## 6. Conclusions

This study demonstrates that ex vivo drug sensitivity correlates with clinical response in 13 pediatric AML patients who received conventional chemotherapy. Additionally, the ex vivo DSP identified the combination of bortezomib/panobinostat as showing greater ex vivo efficacy compared with conventional chemotherapy in children with de novo AML. Azacitidine, in combination with venetoclax, was also identified in the DSP, showing comparable ex vivo efficacy to conventional chemotherapy in the same patient subset. These results support the potential utility of including ex vivo drug sensitivity testing in patient care. Additional studies are warranted to confirm the predictive utility of the DSP in children with AML.

## Figures and Tables

**Figure 1 cancers-14-06240-f001:**
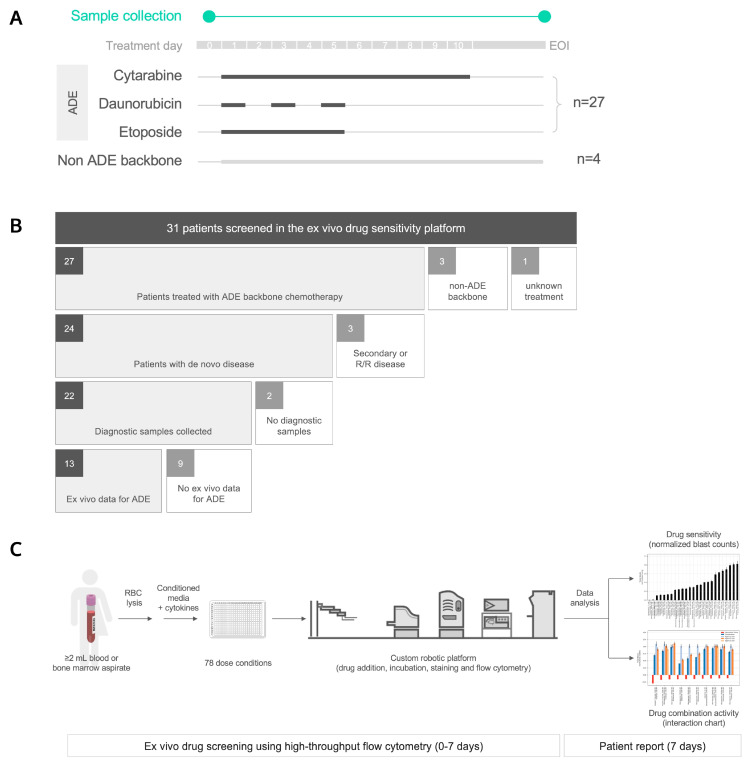
Ex vivo drug sensitivity profiling for 31 pediatric AML patients. (**A**) Fresh blood or bone marrow aspirate samples were collected for this study prior to treatment on day 0, and/or at the end of induction (EOI), from 31 pediatric AML patients diagnosed at Texas Children’s Hospital. Twenty-seven patients received cytarabine, daunorubicin, and etoposide (ADE) as backbone chemotherapy, and four patients did not receive an ADE backbone. (**B**) Patients were stratified by induction chemotherapy, disease status, pre-treatment sample collection, and matching ex vivo data. R/R, relapsed or refractory. (**C**) Ex vivo drug sensitivity was profiled using an automated flow cytometry-based platform. RBC, red blood cell. (**D**) Heat map of ex vivo drug sensitivity results that passed the data quality control, including 48 total samples from 27 pediatric AML patients (columns) in response to 78 dose conditions (rows). Patients were clustered based on differential ex vivo drug sensitivity using hierarchical clustering (Euclidean distance metric, Ward linkage criterion). Cell color indicates normalized blast counts <1 (red; reduction in blasts) or normalized blast counts >= 1 (blue; no or low reduction in blasts). Rows above the heat map indicate selected clinical and biologic variables. UPN, unique patient number; F, female; M, male; R/R, relapse/refractory; A, alive; DD, died of disease; TRM, treatment related mortality; CNS, central nervous system; FAB, French–American–British; BM, bone marrow; PB, peripheral blood; PDX, patient-derived xenograft mouse model; MRD, minimal residual disease; CPX, CPX-351; BMT, bone marrow transplant; Day, readout day for ex vivo drug sensitivity; * status for CPX, ADE, BMT; nan, not reported.

**Figure 2 cancers-14-06240-f002:**
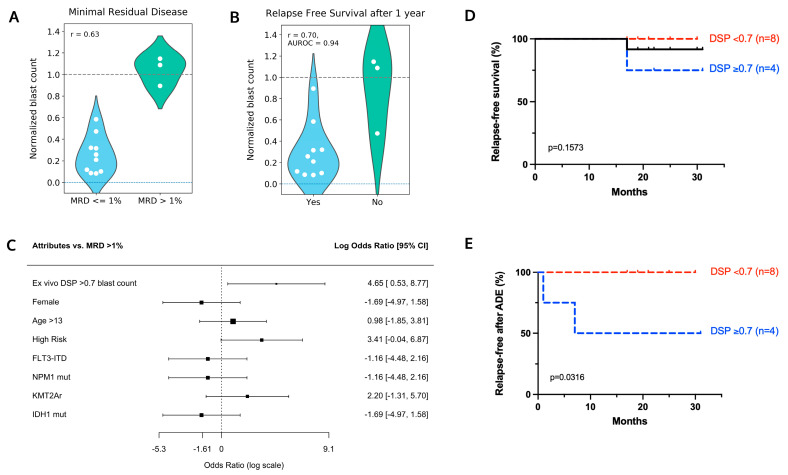
Ex vivo drug sensitivity to ADE correlates with clinical response. Diagnostic samples from 13 de novo pediatric AML patients that received ADE were screened for ADE sensitivity ex vivo. Violin plots to measure the correlation between normalized blast counts in the DSP and (**A**) minimal residual disease and (**B**) one year relapse-free survival. Lower dashed line at y = 0 indicates no blasts remaining following treatment with the drug. Upper dashed line at 1.0 indicates blast counts were comparable to the DMSO control. AUROC, Area Under the Receiver Operating Characteristic curve. (**C**) Log odds ratio comparing MRD > 1% with a single mutational or clinical attribute (rows). Black boxes indicate the log odds ratio on a log scale, with the size of the box being proportional to the number of patient samples. (**D**) Relapse-free survival curves and (**E**) incidence of relapse following induction for 12 patients treated with ADE. One patient died from treatment-related mortality and is excluded from this analysis cohort. Patients were stratified into two DSP groups based on high sensitivity to ADE ex vivo (DSP < 0.7; red) and low sensitivity to ADE ex vivo (DSP >= 0.7; blue). *p*-values were determined using the Log-rank (Mantel-Cox) test.

**Figure 3 cancers-14-06240-f003:**
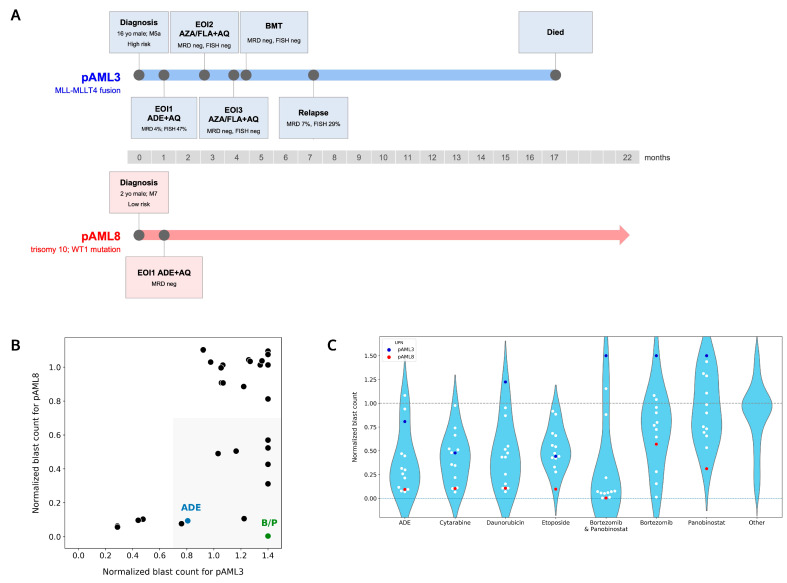
Patients pAML3 and pAML8 show distinct ex vivo drug sensitivity profiles. (**A**) Disease course and treatment timelines for patients pAML3 and pAML8. (**B**) Scatterplot comparing normalized blast counts for patients pAML3 (x-axis) and pAML8 (y-axis) in response to 29 matched drug conditions assayed ex vivo. Shaded region indicates conditions that showed sensitivity for patient pAML8 and not for patient pAML3. B/P = bortezomib/panobinostat. Normalized blast counts > 1.4 are shown as 1.4. (**C**) Violin plot comparing the distribution of normalized blast counts for 13 patients in response to the indicated conditions. “Other” is the mean of the remaining 30 treatment conditions included in the DSP. Patient pAML3 is indicated in blue; patient pAML8 is indicated in red. Lower dashed line at y = 0 indicates no blasts remained following incubation with drug. Upper dashed line at 1.0 indicates no change in blast counts relative to DMSO control. Normalized blast counts > 1.5 are shown as 1.5.

**Figure 4 cancers-14-06240-f004:**
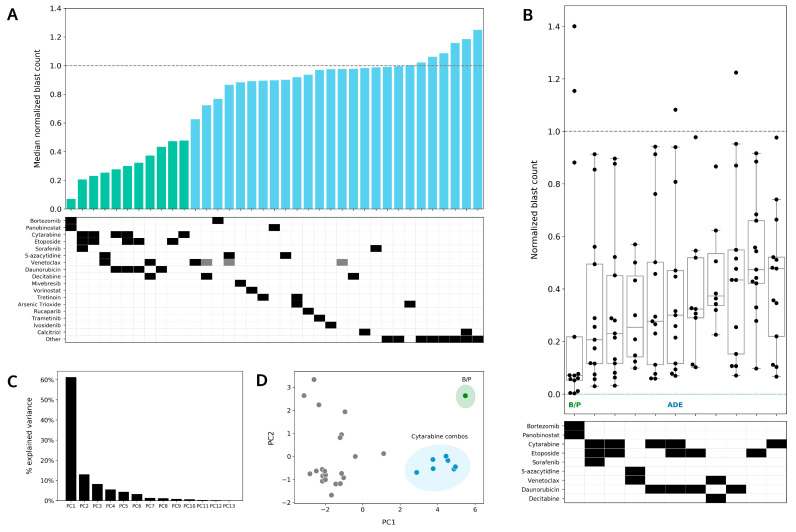
Bortezomib in combination with panobinostat shows greater sensitivity than ADE ex vivo. (**A**) Median normalized blast counts for 13 patients in response to 37 conditions assayed ex vivo. Conditions are ordered from most to least sensitive starting from the left. Green bars highlight the 11 most sensitive conditions in the DSP. The others indicate the following conditions from left to right: enasidenib, gilteritinib, quizartinib, sunitinib, crenolanib, dexamethasone, dexamethasone/calcitrol, and midostaurin. Gray boxes indicate lower drug concentrations. (**B**) Box plot of the top 11 treatments. Normalized blast counts > 1.4 are shown as 1.4. B/P, bortezomib/panobinostat; ADE, cytarabine/daunorubicin/etoposide. (**C**) Percentage of variation in the DSP explained by principal component (PC). (**D**) PC1 vs. PC2 principal component analysis plot to identify clustering of conditions that account for variance in ex vivo drug sensitivity. B/P is shaded in green. Conditions including cytarabine are shaded in blue.

**Figure 5 cancers-14-06240-f005:**
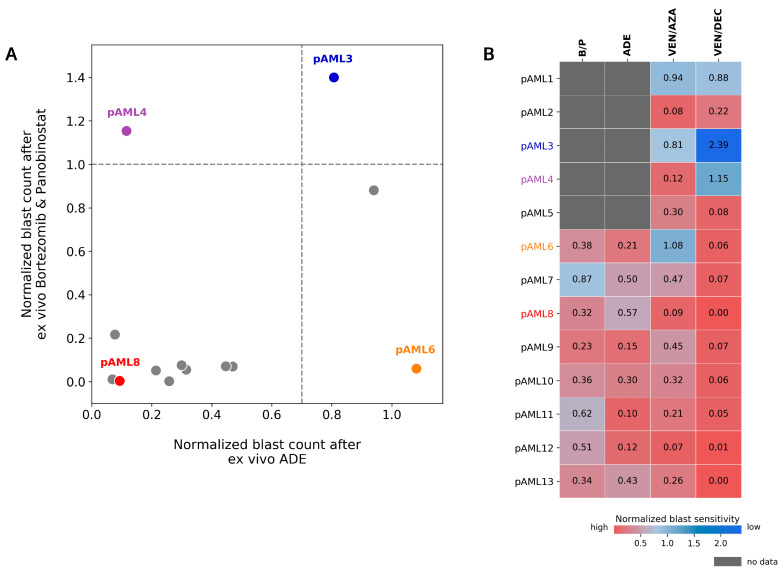
Differential response to ADE and Bortezomib/Panobinostat is observed ex vivo. (**A**) Comparison of normalized blast counts between ADE and bortezomib/panobinostat (B/P) in 13 diagnostic samples from patients with de novo AML. Dashed lines indicate cutoffs for high sensitivity (DSP < 1.0 for B/P; DSP < 0.7 for ADE). DSP values > 1.4 are shown at 1.4. (**B**) Normalized blast counts for B/P, ADE, VEN/AZA (venetoclax/azacitidine), and VEN/DAC (venetoclax/decitabine) on a graded color-scale for high sensitivity (red) and low sensitivity (blue). Grey boxes indicate no data. Patients pAML3 (blue), pAML4 (purple), pAML6 (orange), and pAML8 (red) are labeled in the indicated colors.

**Table 1 cancers-14-06240-t001:** Patient data by UPN. Thirty-one patients were profiled using the ex vivo drug sensitivity platform. Age is shown in years at time of initial diagnosis. Mutations tested: FLT3-ITD, NPM1, CEBPA, and KMT2Ar. Minimal residual disease (MRD) was measured by flow cytometry following the end of induction. Clinical outcome indicates status at last contact. UPN, unique patient number; F, female; M, male; FAB, French-American-British; NR, not reported; ADE, cytarabine/daunorubicin/etoposide; AQ, atovaquone; CPX, CPX-351; AE, cytarabine/etoposide; DA, daunorubicin/cytarabine; DD, death from disease; f/u, follow-up; TRM, treatment-related mortality.

UPN	Age, y; Sex	Disease Status	FAB	Final Risk	Cytogenetics	Mutations	Induction Chemotherapy	MRD, %	Relapse	Clinical Outcome
**pAML1**	15; M	De Novo	M2	High Risk	t(8;21)	Negative	ADE + AQ	9.7	No	Alive
**pAML2**	16; F	De Novo	M2	Low Risk	11q23 dupl	Negative	ADE + AQ	0	No	Alive
**pAML3**	16; M	De Novo	M5a	High Risk	Complex	Negative	ADE + AQ	4	Yes	DD
**pAML4**	7; M	De Novo	M4	High Risk	8	FLT3-ITD	ADE	0	No	Alive
**pAML5**	13; M	De Novo	M2	Low Risk	t(8;21)	Negative	ADE + AQ	0	No	Alive
**pAML6**	11; M	De Novo	M1	High Risk	t(10;11)+ complex	KMT2Ar	ADE + AQ	50	Yes	Alive
**pAML7**	19; F	De Novo	M1	High Risk	Normal	FLT3-ITD; NPM1	ADE + AQ	0.7	No	TRM
**pAML8**	2; M	De Novo	M7	Low Risk	10	Negative	ADE + AQ	0	No	Alive
**pAML9**	16; F	De Novo	M1/M2	Low Risk	Normal	CEBPa	ADE + AQ	0	No	Alive
**pAML10**	16; F	De Novo	M4/M5	Low Risk	Normal	NPM1	ADE	0	No	Alive
**pAML11**	11; M	De Novo	M1/M2	Low Risk	Normal	NPM1	ADE + AQ	0	No	Alive
**pAML12**	12; M	De Novo	M1/M2	Low Risk	Normal	FLT3-ITD; CEBPa	ADE + AQ	0	No	Alive
**pAML13**	6; F	De Novo	M1/M2	Low Risk	t(1;11) not KMT2A, del(11q)	CEBPa	ADE	0	No	Alive
**pAML14**	7; M	Secondary	M5	High Risk	t(11;19p13.1)	FLT3 (2 PMs); KMT2Ar	CPX-351	0	Yes	DD
**pAML15**	13; M	Refractory	M5	High Risk	Normal	FLT3-ITD	ADE +AQ	4.7	Yes	Alive
**pAML16**	unknown	Refractory	unknown	unknown	t(7;21), -17	not done	unknown	unknown	Refractory	lost to f/u
**pAML17**	1; M	De Novo	M5	Low Risk	t(10;11) cryptic	KMT2Ar	ADE	0	Yes	DD
**pAML18**	1; M	Refractory	M7	High Risk	t(1;21)	Negative	ADE + AQ	2	Refractory	DD
**pAML19**	0.75; F	De Novo	M5	Low Risk	inv(16)	Negative	ADE + AQ	1.5	No	Alive
**pAML20**	14; F	De Novo	M1/M2	High Risk	Complex	Negative	CPX-351	4	Refractory	DD
**pAML21**	11; F	De Novo	M5	High Risk	Normal	FLT3-ITD	ADE + AQ	0	No	Alive
**pAML22**	0.83; M	De Novo	M7	High Risk	CBFA1T3-GLIS2	Negative	ADE + AQ	0.02	Yes	Alive
**pAML23**	15; F	De Novo	M2	High Risk	8	FLT3-ITD; NPM1	ADE + AQ	0	No	TRM
**pAML24**	6; F	Relapse	M2	Low Risk	t(8;21), inv(8), del(9q)	Negative	ADE + AQ	0	Yes	Alive
**pAML25**	10; F	De Novo	M7	High Risk	del(7q), +1, +8, der(1;7)	Negative	ADE + AQ	0.23	No	Alive
**pAML26**	14; F	De Novo	M5	High Risk	t(10;11)	KMT2Ar	ADE + AQ	2.5	Yes	Alive
**pAML27**	1; F	Relapse	M4/M5	High Risk	Complex	FLT3-ITD; KMT2Ar	ADEx2, AE, FLAG	0	Yes	DD
**pAML28**	1; M	De Novo	M5	Low Risk	t(9;11)	KMT2Ar	ADE + AQ	0	Yes	Alive
**pAML29**	3; F	De Novo	M5	High Risk	dup(Xq)	Negative	ADE	0	No	Alive
**pAML30**	17; F	Refractory	M7	High Risk	Complex	Negative	ADE	25	Refractory	Alive
**pAML31**	0.58; F	De Novo	M5	Low Risk	t(9;11)	KMT2Ar	DA+gemtuzumab + AQ	0	No	Alive

## Data Availability

For original data, contact Notable Labs at https://notablelabs.com/contact-notable/, accessed on 1 November 2022.

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
