# Peer review of "Ex Vivo Drug Sensitivity Correlates with Clinical Response and Supports Personalized Therapy in Pediatric AML"

_cancers, 2022, doi:10.3390/cancers14246240_

Round 1
Reviewer 1 Report
The authors investigated ex vivo drug sensitivity on AML cells and compared clinical response in pediatric AML. The data seems to relatively precious and the manuscript is well organized though sample size is small. But, drug sensitivity in vivo may be affected by many factors including ex vivo drug sensitivities. Therefore, the authors should add more discussion including recent similar articles about drug sensitivity for leukemia including ex vivo test .
Author Response
Point 1: Therefore, the authors should add more discussion, including recent similar articles about drug sensitivity for leukemia including ex vivo test.
Response 1: We and others have demonstrated the feasibility and clinical utility of functional precision medicine assays in hematologic malignancies. Discussion in the Introduction section on ex vivo drug sensitivity testing has been updated to include two recent publications showing clinical utility of an ex vivo drug sensitivity test in leukemia:
- Zhang, Y.; Ji, M.; Zhao, J.Y.; Wang, H.F.; Wang, C.W.; Li, W.; Ye, J.J.; Lu, F.; Lin, L.H.; Gao, Y.T.; et al. Ex Vivo Chemosensitivity Profiling of Acute Myeloid Leukemia and Its Correlation With Clinical Response and Outcome to Chemotherapy. Frontiers in oncology 2021, 11, 793773, doi:10.3389/fonc.2021.793773.
- Lin, L.; Tong, Y.; Straube, J.; Zhao, J.; Gao, Y.; Bai, P.; Li, J.; Wang, J.; Wang, H.; Wang, X.; et al. Ex-vivo drug testing predicts chemosensitivity in acute myeloid leukemia. J Leukoc Biol 2020, 107, 859-870. doi:10.1002/JLB.5A0220-676RR.
Reviewer 2 Report
The authors report ex vivo drug sensitivity correlates with clinical response and supports personalized therapy in pediatric AML.
1. It was not clear why the authors selected bortezomib in combination with panabinostat in this study. Treatment dose, timing and duration was not provided. The authors should clarify these things.
2. It was not clear the mechanism of bortezomib in combination with panabinostat in this study. The authors should provide the signaling data and describe in detail.
Author Response
Point 1: It was not clear why the authors selected bortezomib in combination with panabinostat in this study. Treatment dose, timing and duration was not provided. The authors should clarify these things.
Response 1: The combination of bortezomib and panobinostat was included in this study as part of a broad panel of FDA-approved drugs that Notable uses for ex vivo drug sensitivity screening in leukemia patients (Fig 1D). Across all the conditions tested, bortezomib in combination with panobinostat showed the highest sensitivity in the majority of patient samples screened (Fig 3C, 4B). Drug concentrations (treatment dose) are shown in Fig 1D, timing and duration for all conditions tested in the ex vivo drug sensitivity platform were the same; all drugs were added on Day 0 and changes in blast populations were read out on Day 3 and/or Day 7 depending on sample size. The Methods section has been revised as follows to make this more clear:
Before
Ex vivo drug sensitivity platform (Methods section)
Ex vivo drug sensitivity testing was performed at Notable Labs (Foster City, CA), using a custom robotic flow cytometry platform. Within 72 hours of collection, fresh blood or bone marrow aspirate samples were incubated with 1X RBC lysis buffer (eBioscience) to remove red blood cells. Cell pellets were resuspended in StemSpan Serum-Free Expansion Medium II (SFEMII; STEMCELL Technologies) containing HS-5 conditioned medium and Penicillin/Streptomycin (Corning) with the following cytokines: rhIL-7, rhG-CSF, rhFLT3L, rhSCF, rhTPO (Miltenyi Biotec); rhGM-CSF, rhIL-3, rhIL-6 (R&D Systems). Resuspended cells were transferred to 384-well plates at 15,000 cells per well and incubated with drug in triplicate or DMSO control in sextuplicate. Changes in tumor blast populations were assayed by flow cytometry on days 3 and 7 following addition of drug. Patient samples with low total counts were assayed at a single time point on day 3 or day 7.
Revised
Ex vivo drug sensitivity platform (Methods section)
Ex vivo drug sensitivity testing was performed at Notable Labs (Foster City, CA), using a custom robotic flow cytometry platform. Within 72 hours of collection, fresh blood or bone marrow aspirate samples were incubated with 1X RBC lysis buffer (eBioscience) to remove red blood cells. Cell pellets were resuspended in StemSpan Serum-Free Expansion Medium II (SFEMII; STEMCELL Technologies) containing HS-5 conditioned medium and Penicillin/Streptomycin (Corning) with the following cytokines: rhIL-7, rhG-CSF, rhFLT3L, rhSCF, rhTPO (Miltenyi Biotec); rhGM-CSF, rhIL-3, rhIL-6 (R&D Systems). Resuspended cells were transferred to 384-well plates at 15,000 cells per well and incubated with drug in triplicate or DMSO control in sextuplicate (day 0). After 3 or 7 days following addition of drug, changes in tumor blast populations were assayed by flow cytometry. Patient samples with low total counts were assayed at a single time point on day 3 or day 7.
----
Point 2: It was not clear the mechanism of bortezomib in combination with panabinostat in this study. The authors should provide the signaling data and describe in detail.
Response 2: While we recognize the importance of characterizing drug mechanisms, the ex vivo drug sensitivity platform is designed to identify different leukemic blast populations and does not provide signaling data. Given the limited sample size of patients, we could not include multiple staining panels or other assays that could provide mechanistic data in this study. We are currently working on expanding the ex vivo drug sensitivity platform to include signaling data for future studies.